# Change in the Constricted Airway in Patients after Clear Aligner Treatment: A Retrospective Study

**DOI:** 10.3390/diagnostics12092201

**Published:** 2022-09-11

**Authors:** Georgia Fountoulaki, Andrej Thurzo

**Affiliations:** 1Department of Stomatology and Maxillofacial Surgery, Faculty of Medicine, Comenius University in Bratislava, 81250 Bratislava, Slovakia; 2Department of Simulation and Virtual Medical Education, Faculty of Medicine, Comenius University in Bratislava, Sasinkova 4, 81272 Bratislava, Slovakia

**Keywords:** orthodontics, airway, clear aligners, 3D diagnostics, sleep apnea, CBCT

## Abstract

This retrospective study evaluated changes in the pharyngeal portion of the upper airway in patients with constricted and normal airways treated with clear aligners (Invisalign, Align). Additionally, we assessed the change of tongue position in the oral cavity from a lateral view. Evaluation was performed with specialized software (Invivo 6.0, Anatomage) on pretreatment and post-treatment pairs of cone beam computed tomography imaging (CBCT) data. The level of airway constriction, volume, cross-section minimal area and tongue profile were evaluated. Patients with malocclusion, with pair or initial and finishing CBCT and without significant weight change between the scans, treated with Invisalign clear aligners were distributed into two groups. Group A consisted of fifty-five patients with orthodontic malocclusion and constricted upper airway. Control group B consisted of thirty-one patients with orthodontic malocclusions without any airway constriction. In the group with airway constriction there was a statistically significant increase in volume during therapy (*p* < 0.001). The surface of the most constricted cross-section of the airway did not change significantly after treatment in any of the groups. The final tongue position was different from the initial position in 62.2% of all clear aligner treatments. The position of the smallest clearance of the airway in the pharynx was similar for both groups localized at the level of 2nd cervical vertebra.

## 1. Introduction

As imaging techniques become more advanced, radiologic evaluation of various head and neck disorders is also reaching new heights. In recent decades, lateral cephalometry has played an important role in the analysis of the upper airway. However, the morphology of three-dimensional airways cannot be properly represented by two-dimensional analysis [1]. The legacy of 2D lateral cephalographic X-ray, posterior-anterior cephalograms and panoramic radiographs has led to 3D imaging, overcoming low resolution and imaging errors allowing calculation of the cross-sectional areas of the airway [2], elevating the precision of treatment of patients with orthodontic malocclusions, and assisting specialists to perform with highest details in their treatment plan [3,4].

Cone beam Computed Tomography (CBCT) imaging has been utilized in scientific practice [5,6] in dentistry to obtain three-dimensional (3D) images of patients and in medical education for nearly fifteen years [7] orthodontics it provides information about teeth malposition represented as divergencies on the x-y-z planes of space, which are coronal, sagittal, and axial [8]. Additional parameters visualized during CBCT imaging are articulation, temporomandibular joint, intermaxillary relationships, evaluation of the morphometric properties in maxillofacial regions, and precise visualization of head and neck anatomy including the upper airway morphology with its possible constrictions. For soft tissue morphology magnetic resonance imaging is frequently employed [9].

Studies conducting dimensional evaluation of the upper airway have proven reliable [10,11] and point out that breathing through upper airways is of great importance for normal craniofacial development [12]. During facial growth, alterations in upper airway breathing can affect the stomatognathic system and the development of the structures as well as their functions [13,14].

Research published by Tsolakis in 2016 concluded that CBCT is an accurate method for measuring anterior nasal volume, pharyngeal volume, and pharyngeal minimal cross-sectional area [15].

Mild forms of air cessation during sleep, obstruction of the upper airway, and even muscle tone loss [16] can significantly affect quality of sleep. Since undisturbed breathing patterns and REM sleep is now better understood [17,18] the crucial REM phase has been linked with various brain disorders [19,20], as the upper airway is stiffer and less compliant during REM sleep than during NREM sleep [21]. Upper airway constriction is a multifactorial problem that can result from hypertrophic adenoids, a retrognathic mandible, atrophy of suprahyoid muscles especially lateral pterygoid and genioglossus muscle [22], obesity, as well as many other factors. Obstructive sleep apnea (OSA) is a common sleep disorder represented by repetitive nocturnal upper airway collapse accompanied by intermittent hypoxia, fragmented sleep, fluctuations in blood pressure and highly active sympathetic nervous system activity [23,24]. Sleep questionnaires and the number of apnea/hypopnea incidents per night determine the level of severity of OSA, measured by the apnea/hypopnea index (AHI) [25,26] assessed during Polysomnography (PSG), which is an important but costly and time-consuming tool for identifying OSA. In 2011, a Spanish team consisting of Dr. Fortuna used breath analysis to find that fractional exhaled nitric oxide (a vasodilator), is higher in OSA patients, assisting in identification, treatment, and monitoring of the patients under mechanical ventilation. A meta-analysis performed by Armalaite J. concluded that the reduced upper posterior pharyngeal space (SPAS) could be a prognostic parameter for suspected OSA, while the mandibular plane to the hyoid bone (MP-H) could be used as a predictor when differentiating normal subjects and patients with OSA [27].

OSA affects more than 50% of the adult population; in non-communicable disease patients, 27.9% males and 23.8% females [28] in patients with cardiovascular disease 50% to 80%, while in half of the subjects with heart failure the mortality and the prognosis are worse [29]. In the medical field Alexandropoulou confirmed that the 27.7% of the staff of the 444 Greek nursing staff population experienced daytime sleepiness [30].

The signs of OSA include snoring while sleeping, unrested sleep, narcolepsy, cognitive impairment, mood elevations, personality changes and daytime sleepiness, with the latter being the link between OSA and depressive symptoms as investigated by Celik Y. [31]. In a population of children with snoring that was investigated by Hsieh Hui-Shan, screening consisting of home sleep pulse oximetry combined with adenoidal-nasopharyngeal ratio (ANR) showed higher effectiveness than ANR and tonsil size [32,33]. Furthermore, there was positive correlation between the accident statuses of drivers with OSAS severity based on data acquired in a study by Celikhisar H. [34].

Orthodontists, with their understanding of malocclusions in the context of the intermaxillary relationship, tongue posture, airway, and bone morphology, can build customized treatment plans for their patients. Frequently, patients are coached during aligner therapy with a smart mobile app, as their treatment compliance is essential, but also for reporting possible sleep disruptions or other patient feedback, providing valuable research data [35].

Very often, Anterior-Posterior (AP) discrepancy is an outcome of an underlying skeletal discrepancy best dealt with in growing patients with growth modification. In clear aligner technology the mandibular advancement feature can be used to address Class II skeletal discrepancies in growing patients with mandibular retrognathia. In Class III patterns that are characterized by maxillary retrusion, growth modification with a protraction face mask can be attempted first, and after Class I has been established the treatment can be followed by clear aligners.

In cases where the AP discrepancy is mild, dental camouflage can be used in combination with elastics wear and/or premolar extraction. In severe cases, where dental camouflage is not possible, orthognathic surgery is required for correction of the underlying skeletal problem. From modern approaches to correct dental Class II a 3D printed distalizer can be utilized [36].

Understanding the biomechanics of the clear aligner technique, the ability to digitally plan treatment, and predictable treatment protocols, are keys to the success of any clear aligner treatment.

The purpose of this study is to present a concept and method for 3D evaluation of upper airway morphology and tongue position using initial and final CBCT images that accompany orthodontic treatment with clear aligners. This work compares the clinical effects of aligner therapy on upper airway morphology in patients with stenosis before treatment and in patients without stenosis. The negative effects of upper airway narrowing on sleep quality and cognitive function in humans are well known. This study sheds light on the relationship between orthodontic correction of dental malocclusions and airway morphology, which has been poorly studied scientifically.

## 2. Materials and Methods

### 2.1. Concept, Hypothesis and PICO

Pre-treatment and post-treatment airways can be 3D-evaluated on CBCT diagnostic records made during orthodontic aligner treatment. Segmentation and evaluation of airway morphology from CBCT is done with specialized software, and the tongue position from CBCT is evaluated by an orthodontist.

The clinical effect of corrected occlusion and/or intermaxillary relationship can influence the upper airway morphology in the oropharyngeal portion of the pharynx (Figure 1).

The working hypothesis of this paper was a presumption that if the pre-treatment constriction of airway is an outcome of disturbed occlusion and/or incorrect intermaxillary relationship, an orthodontic correction would result in characteristic improvement of this constricted airway morphology, especially in comparison to an unconstricted control group treated with the same methods.

The null hypothesis presumes that if orthodontic aligner treatment has no effect on the upper airway morphology and tongue position, there shall be no significant difference on upper airway morphology between pre-and post-treatment of non-growing patients.

The PICO framework of this paper is the following:Patients were 55 adults with malocclusion with initial and final CBCT, without significant BMI change between the CBCT scans and with observed airway constriction on the pre-treatment CBCT.Treatment with clear aligners (Invisalign).The control group and 31 patients with malocclusion without any airway constriction were treated with the same approach.The outcome of the treatment was a hypothetical difference in airway and tongue position that was evaluated.

### 2.2. Borders of the Airway

The upper border of the airway was defined by the diagonal line connecting the anatomical point of Sella Turcica and the posterior nasal spine (PNS). Sella turcica of pituitary (hypophyseal) fossa is a midline, dense structure in the sphenoid bone, which houses the pituitary gland. The posterior nasal spine represents the most posterior point of the maxillary line and nasal floor seen on the lateral projection of the CBCT of the patient (Figure 2).

The lower border was defined as the base of the epiglottis, the soft tissue that leads air further into the lungs, dividing the upper respiratory system from the lower, as well as the respiratory system from the digestive system. The borders were defined by the study by dos Santos et al. in 2020 during their trial to correlate airway volume and maximum constriction area location in different dentofacial deformities [1].

The level of constriction was determined by the level of the cervical vertebrae (CV), again using the CBCT of the patient in the section of airway in the software Invivo Anatomage. The cervical vertebrae can be divided into three parts, the upper border, the middle, and the lower border. Each of the cervical vertebrae can be divided into these three levels and the constriction of the airway can be defined accordingly. The most common borders where airway constriction is found are the middle border of CV2, the upper border of CV3 the Middle of CV3 as well as the lower border of CV3.

When the software analyzes the input data and evaluates the airway, different colors of the color spectrum illustrate the constriction of the airway. Severe airway obstructions are represented by black, dark red, red followed by orange, yellow and green. The green color categorizes the airway into adequate passage of air followed by blue and white representing the widest airway dimension.

### 2.3. Group Selection and Clinical Evaluation

Only adult non-growing patients were selected without any significant BMI change between an average two-year span between CBCT scans. The first group consisted of patients with constricted airway identified at the CBCT 3D analysis, while the second group included patients with adequate airways without significant difficulties or abnormalities in their breathing pattern during the day or during sleep. In total, the first group numbered fifty-five patients with constriction of their airways, while the second group was the control group consisting of thirty-one patients.

An important criterion was the CBCT scan of the skull showing all the skeletal craniofacial structures of the head and neck necessary for the analysis. The initial, as well as the final CBCT after treatment, were necessary for the research. All CBCT scans were evaluated by an orthodontist within complex orthodontic treatment planning or as pre-finish evaluation due to relevant clinical reasons.

Example of the software 3D evaluation of the airway from the CBCT shows pre-treatment airway constriction (Figure 3a) and post-treatment evaluation after orthodontic therapy (Figure 3b). This example shows rather insignificant improvement of airway constriction as well no significant change of tongue posture (blue).

Evaluation of the tongue position was added to the methodology to objectivize the airway changes, as these may be affected by different tongue positions between initial and final CBCT scans. The tongue can affect the airway clearance.

Research from 2016 published by Pliska et al. in the American Journal of Orthodontics and Dentofacial Orthopedics was focused on the effect of orthodontic treatment on the upper airway volume in adults. In this retrospective study, authors examined the effects of orthodontic treatment with and without extractions on airway clearance. The Wilcoxon signed rank test was used to compare volumetric and minimal cross-sectional area changes from pretreatment to post treatment. Conclusions of this research were that “dental extractions in conjunction with orthodontic treatment have a negligible effect on the upper airway in adults” [37].

By contrast, research conducted by Wang 2012, an orthodontist researching changes of pharyngeal airway size and hyoid bone position following orthodontic treatment of Class I bimaxillary protrusion, concluded that a significant correlation existed between the retraction distance of lower incisor and the airway behind the soft palate, uvula, and tongue. Wang et al. also stated that “the pharyngeal airway size became narrower after the treatment. Extraction of four premolars with retraction of incisors did affect velopharyngeal, glossopharyngeal, hypopharyngeal, and hyoid position in bimaxillary protrusive adult patients” [38].

In this study, two independent clinicians evaluated changes of the tongue position in CBCT lateral view in relationship to the surrounding oral cavity. Evaluation of the change of the tongue position in the oral cavity was binary (Yes/No). The upper contour of the dorsum linguae was identified (Figure 4) and compared between initial and final scans. Only significant changes of the contour of dorsum linguae were evaluated as “different posture”. The clinicians were unable to recognize if the recorded position from the final CBCT was the true relaxed position of the patient’s tongue or just a temporary situation during final CBCT scanning.

Standardization and calibration are necessary in research that relies on clinical examiners to subjectively assess typically visual variables. All major objectives in this paper were assessed by specialized software, except for the simple binary task of tongue position change evaluation.

Intra-rater reliability was estimated by having the rater read and evaluate each tongue position two times in two separate dates. As the intra-examiner reproducibility was 100% in this simple task, all further elaborated examiner training was waived.

Examiners agreement in Group A was 54/55 (98.20%) and in the Control group was 31/31 (100%). In the only case of initial disagreement, a mutual agreement was reached. Inter-examiner reliability—reaching agreement between two examiners—was typically more tedious, although only one case of disagreement occurred. For expressing inter-examiner reliability, the kappa statistic with examiner calibration protocols was considered not necessary to a wider extent than already stated in the statistical evaluation provided.

Various clinical situations were evaluated. In situations where the contour of the dorsum linguae could not be differentiated from surrounding soft tissues it was considered in continuous contact. If such continuous contact was on the initial and as well final CBCT scan, tongue position in oral cavity was evaluated as unchanged, although the treatment could have changed teeth positions in the alveoli, and thus oral cavity shape and physical borders for tongue could have changed (Figure 4 top). Figure 4 shows three distinct examples of clinical situations where a not necessarily worsened position of tongue posture results in worsening of airway constriction (Figure 4 below), and post-treatment change of tongue position in the oral cavity (Figure 4 middle) improving an even previously unconstructed airway.

### 2.4. CBCT and Instructions to the Patients

All the CT studies were performed by radiologists/assistants in a private clinic in Bratislava using the same CBCT (CT i-CAT Imaging Sciences International, Hatfield, PA, USA). The scanning protocol was 120 kV, 36.9 mA, 13 × 23 cm field of view, 0.3-mm voxel.

Before scanning, the patients were instructed not to move their head, nor to swallow. Furthermore, they were asked to have maximum occlusion/inter-cuspidation to avoid discrepancy and reduce the variation [39].

### 2.5. Software Analysis. Invivo 6.0 Anatomage

Two observers evaluated all the CBCT scans of all patients before and after the treatment independently and analyzed them using InvivoAnatomage software (version 6.0 Imaging, Santa Lara, CA, USA). The initial and final Dicom (Digital Imaging and Communications in Medicine) images were imported into the software, observed, and evaluated. Each analysis was stored in a separate file in native *.inv format.

While using the software, the position of the head of each patient was corrected using the 3D orientation widget and parameters such as the Frankfurt horizontal line, superimposition of the skeletal base of the skull in the level of Sella Turcica, and bony maxilla.

The observers performed the analysis by selecting and defining anatomical points and skull characteristics using Invivo in the section of 3D analysis.

When the selection was completed, the software automatically calculated the results which were numerical values and angles. The data of the analysis before and after the treatment were gathered in a table that is provided as supplementary material to this paper.

The table includes pseudonymous identification of the patient, the airway data analysis as well as skeletal hard and soft tissue analysis. The personal data of each patient consisted of the name, the surname, the gender, the starting age, starting year and the year that the patient finished their treatment. All personal information was removed after data collection was completed and graphical data were anonymized. In the final table, the portion of airway analysis calculated the volume of the airway within defined boundaries, the minimum area of the most constricted section, and the level of the maximum constriction before and after the treatment. The level of the airway constriction was determined by skeletal cervical vertebrae.

### 2.6. Aditional Parameters Collected and Evaluated

The data collected from both groups, the one with constricted airways and the control group, were gathered in one complete Microsoft Excel file (Microsoft Corporation, Redmond, WA, USA). The file consisted of columns representing the skeletal information obtained from the CBCT analysis as well as information about the treatment with clear aligners and the status of the airway before and after the completion of the treatment. The patient answers from clinical examination were recorded in the patient journal and transferred to the final table in a format suitable for the statistical software processing.

Photographic documentation and intraoral scanning of the dental arches of the patient provided information about the initial dental status and dental relationship including the Angle class categorized in 3 classifications, Class I, II and III. Initial Class and final Class were taken into consideration.

Other data assessed in the table were the total airway volume defined in cc, minimal constriction area in mm^2^ and the level of the constriction of the airway according to level of cervical vertebrae. Invivo 6 software provides visualization of the airway as well as calculating the volume of the air. Selected points across the airway provided the volumetric result, adjusting for both the upper and the lower border of the airway.

Parameters used during the treatment such as the use of intermaxillary (between upper and lower arch) elastics, or a special feature of the design of aligners called wings that assist the forward step movement of the mandible, were taken into further consideration.

Cephalometric parameters, as well as the definition and significance of their content, are shown in the Table 1.

The position of the tongue was observed in both initial and final CBCT analysis and, in the case of difference in posture, the change was noted.

Using Invivo 6.0 Anatomage, the orthodontist could analyze, segment, and export the airway in a special format named a Standard Tessellation Language (STL) file and, using 3D printing, could expand the possibilities of additional analysis.

### 2.7. Data Collection and Statistical Analysis

Group A was defined as patients treated with clear aligner therapy (Invisalign) with upper airway constriction prior to the treatment.

Control group B represented a set of patients treated from the initial stage without identified airway constriction, based upon CBCT 3D analysis.

Patients were adults without significant weight change (±4%) between the initial and final scan.

All 3D comparative analyses of the CBCTs before and after treatment were performed in the years 2021–2022. Work on this study started in 2017. Data were evaluated by professional statisticians acknowledged at the end of this paper. Data were statistically analyzed using SPSS 23.0 software (Chicago, IL, USA) and GraphPad Prism 6.01 (La Jolla, CA, USA). The threshold of statistical significance was set to *p* < 0.05.

### 2.8. Aditional Analysis

Additional helpful diagnostic tools were exported from 3D software (Invivo) such as panoramic and cephalometric X-ray, temporo-mandibular joint (TMJ) visualization, inferior alveolar nerve tracking, differential map of hard and soft tissues asymmetries pre-treatment and post-treatment by creating a half mirror image according to one side of the patient’s face. Information such as this introduced the Golden ratio of soft tissues, and made the orthodontic treatment more soft tissues-driven. Three dimensional dental casts and 3D extra-oral face-scans were imported into the software and were adjusted to the skeletal and soft tissues of the patient, allowing further analysis.

## 3. Results

### 3.1. Clinical Structure of the Selected Dataset

Of 120 consecutive patients, 86 fit the selection criteria. All selected cases were non-extraction treatments. The most frequent reason for exclusion was significant change of their BMIs` during and after illness. From the selected, 64% were identified with some form of upper airway constriction on the initial CBCT, and the 36% who had adequate airways was defined as control group. Of all included treatments in this total, 72.1% were female and 27.9% male with an average treatment time of 2.1 years.

Intermaxillary elastics were used in 61.8% in group A and 74.2% in group B (control). Wings were used in 5.5% of group A and 6.5% (control).

In addition, 5.5% of group A and 3.2% of group B (control) underwent orthognathic intervention. Only Class II mandibular advancement mono-maxillary surgeries were in the sample.

Initial malocclusions were divided into classes according to Angle defined by the position of canines and molars. The distribution was:

20.9% Class I,

66.3% Class II*,

12.8% Class III*.

While at the end of the treatment with clear aligners the percentages were:

65.1% Class I,

32.6% Class II*,

2.3% Class III*.

* As Class II and Class III were considered even unilateral or extraction cases upon position of first molars.

Group A and the control group did not differ statistically significantly in the proportion of women and men (Figure 5a). Age distribution in the groups did not differ statistically significantly (Figure 5b).

In analysis of the application of elastics, wings, and surgery interventions, there was no statistically significant difference between groups.

Neither initial nor final distribution of patients’ Angle Classifications was statistically significant between the groups. As Figure 6a shows, the dominant group in patients with constricted airways were patients in Class II. However, Class II patients were also dominant in the unconstricted control group After finishing CBCT, the majority in both groups were patients in the first Angle Class, although Class II persisted due to strict evaluation of molar position. This included extraction cases, unilateral Class II and non-surgical camouflage treatments in adults (Figure 6b).

### 3.2. Evaluation of the Changes in the Airway


Volume (mm^3^),Minimal Area(mm^2^),Level of constriction.


#### 3.2.1. Volumetric Changes in the Airway

Concerning volume of the airway before the treatment, group A had a value of 19.37 cm^3^ with standard deviation of 5.15 and after the treatment 22.29 cm^3^, while the control group had 33.18 cm^3^ initially and after treatment the value was 32.25 cm^3^. As Figure 7a (pretreatment) and Figure 7b (post treatment) show, the aligner treatment improved the volume in Group A (with pretreatment airway constriction), although resulted in a slight insignificant decrease of airway in the control group. In the group with airway constriction, there was a statistically significant increase in volume during therapy (*p* < 0.001).

#### 3.2.2. Surface of the Airway Cross-Section at the Level of Highest Constriction

There was a significant increase of 16.69 mm^2^ at the mean value of the cross-sectional minimal area of the airway in the group with constricted airway after the treatment with clear aligners. In the control group there was a reduction of 13,4 mm^2^.At the beginning, the minimum area in the control group was statistically significantly larger (*p* < 0.001). At the end of the aligner therapy, the minimum area in the control group remained statistically significantly larger (*p* < 0.001) compared to the group with constriction.

Before treatment, the mean value of the minimal cross-section airway was 130.28 mm^2^ versus 146.97 mm^2^ after treatment, as shown in Figure 8.

The Wilcoxon Signed Ranks test showed 36 positive ranks and 19 negatives out of the 55 patients, meaning that the minimal area at the end of the treatment was greater in 36 cases. The mean value at the beginning for the control group was 246.68 mm^2^ versus 233.28 mm^2^ after the end of the orthodontic treatment.

#### 3.2.3. Position of the Highest Constriction

The locality of constriction in all 86 cases was assessed. Figure 9 shows the localizations before and after the treatment. The most common constriction was identified at the level of the 2nd cervical vertebrae (CV2) in the oropharyngeal portion of the pharynx. After the treatment with Clear Aligners, the most persistent constriction level, the Middle of CV2, reached even higher values but the rest had a reduction. There was no statistically significant difference between the groups.

### 3.3. Change of Tongue Position in the Oral Cavity from Lateral View

The dorsum of the tongue (dorsum linguæ) was evaluated as the profile contour of tongue position on the CBCT. The final contour was compared to the contour from the initial CBCT. Observed differences are shown in different positions of the tongue in Figure 3 and Figure 4. Aligner treatment in patients with constriction on the initial CBCT resulted in a changed position of the tongue in 61.1% cases, and similarly 63.3% in the control group (Figure 10).

### 3.4. Results of Skeletal Analysis

Skeletal analysis evaluating skeletal (hard tissue) parameters in Group A in SNA and SNB found no significant statistical differences. SNA and SNB are angles formed by Sella Turcica, which is a bony depression in the sphenoid bone in the skull, hard tissue nasion which is the most anterior point of frontonasal suture and point A and point B at maximum convexities of maxilla and mandible, respectively. In group B, there was a statistical difference between SNB2 (*p* = 0.031) and SNB2 (*p* = 0.059), as presented in Figure 11.

Another parameter that was taken into consideration was ANB, an angle formed by the skeletal points A and B and the point Nasion, but no statistical difference was found. A dental parameter that was also examined was U1SN, representing the inclination of the upper central incisors to the SN (Sella-Nasion) line with a mean value of 99.40 before and 97.81 after treatment in group A, and 105.74 and 100.12 in control group B, respectively.

A comparison of the values of the lower central incisors’ inclination in reference to the cephalographic lines NB is presented in Table 2 for both groups.

The overjet represents the horizontal distance between the upper and the lower dental arch and their comparison before and after the treatment in group A with clear aligners.

OJ1 represents the initial overjet while OJ2 is the after-treatment overjet as shown in Table 3. Similar results were shown in Group B (control) when the mean value OJ1 was 4.08 mm versus 3.33 mm. The parameters concerning the gonial angle and the relationship of the skeletal bite were not significant.

## 4. Discussion

Results of this study confirmed its working hypothesis that improvement of the occlusion and/or intermaxillary relationship results in improvement of airway clearance, especially in cases where prior constriction was present. In simplified approximation, the upper airway constriction was present in proportion AI:AII:AIII as 3:7:1. AII was also dominant in the group without airway constriction.

Results also identified the most frequent location of upper airway constriction in the level of the second cervical vertebrae. An unconfirmed hypothesis is that this constriction could be most prevalent at this level due to the hypertrophy of the adenoids/tonsils.

On average, every second patient had a different posture of the tongue at the final CBCT. There was no significant difference in this finding between either group. It is impossible to prove the causal nexus of tongue posture, treatment, and airway constriction by our research. Nevertheless, it would be a valuable scientific contribution to identify an established link between the most probable cause and its resulting effects. In some cases, an upper arch expansion and mandibular advancement led to a less constricted airway. Better understanding of the link between tongue position and airway clearance is necessary. Tongue position shall also be evaluated in transversal direction.

A recent study by Lin et al. 2022, demonstrated that hypertensive patients tend to have larger upper airway length, smaller total airway volume and smaller width of upper airway, predisposing them to the risk of developing Obstructive Sleep Apnea (OSA). As the prevalence of OSA is increasing, dental practitioners should be equipped with basic knowledge of OSA [40].

The null hypothesis of this paper was rejected as results proved that most of the treatments resulted in positive volumetric change of the airway, although no significant change of the area in the point of most prominent constriction.

CBCT imaging is a valuable tool that assists in localization of constrictions of the upper airway, but is restricted during the time of exposure, creating limitation of precise evaluation of the airway. The amount of increase and decrease of the volume of the airway during inhalation and exhalation cannot be precisely captured during the exposure of the CT. The total volume of the airway, the minimal area and maximum constriction can have different values during the process of breathing. Antosz et al. 2015, comment on “the flaw of using CBCT to measure airway changes in subjects with sleep apnea due to the fact that the CBCT is static and could vary within the same patient from one day to the next and from one moment to the next” [41].

CBCT does not function as the sole tool for diagnostics for obstructive sleep apnea, and special sleep studies, oximetry and other measurements are necessary to diagnose OSA. CBCT of the maxillofacial region frequently reveals a high percentage of clinically relevant additional findings [42]. In general, there is a lack of evidence related to 3D pharyngeal airway space changes after orthodontic treatment. The largest effect in adults characterized by mandibular retrognathism is orthognathic treatment with mandibular advancement surgery, achieved by means of bilateral sagittal split osteotomy (BSSO) advancement surgery. This intervention leads to a significant, immediate increase in total airway volume, and minimum constriction area. All these changes remained stable at a one-year follow-up [43]. Recently, various digital methods have been introduced for quantification of airway changes suitable for retrospective studies [44].

This paper does not present any special quality of aligner orthodontic treatment in comparison to other orthodontic techniques, but rather clarifies a uniform technique in the treatment. Either way there is a plethora of different treatment modalities in every orthodontic technique that makes the clinical approach difficult to compare. Neither clinical conditions and extent of patient malocclusion, nor compliance, are scientifically comparable. The effect of clear aligners includes expansion of the hard palate in the form of crown tipping and body teeth movement buccally. In some cases, skeletal anchorage or elastics were involved. Evidence confirming that pure dental expansion could cause remodeling of the upper airway is not available.

A recent retrospective study by Diwakar et al. 2021, focused on the effect of craniofacial morphology on pharyngeal airway volume calculated using CBCT. An important finding of this study is that pharyngeal airway space differs significantly between males and females [45].

Studies have been made to assess dimensional changes in the upper airway after appliance or surgical therapy in patients with OSA, and to find correlation of CBCT findings with treatment outcome. There is evidence that CBCT-measured anatomic airway changes with surgery and dental appliance treatment [46] but there is insufficient literature pertaining to the use of CBCT to assess treatment outcomes to reach a conclusion [4].

Furthermore, a study of reliable volumetric assessment of the nasal airway by Mupparapu et al. 2021, concluded that there is lack of a gold standard, but it is possible to quantify nasal airway volume and its reduction [47].

Limitations of this study include the limitation of patient selection due to body mass index change that seemed more frequent in patients with CBCT prior to 2020 and after onset of the COVID-19 pandemic. A CBCT scan is particularly justified when it brings a clear benefit to the patient’s treatment when compared with conventional imaging techniques, and CBCT should be considered for clinical orthodontics for selected patients [48,49].

Another limitation of this study is the fact that the pharyngeal airway space differs significantly between males and females, and that head posture in relation to the neck when standing in the CBCT can deform the airway; although the head position is repositioned, airway deformation can remain. Similarly, a limitation is unnatural tongue spasm during CBCT in unrelaxed patients, which is difficult to identify, as well as the necessity to evaluate tongue position in lateral directions.

Large clinical heterogeneity of the sample can be considered a limitation for the interpretation of the findings.

## 5. Conclusions

Patients who underwent clear aligner therapy experienced changes in upper airway volume. In the airway narrowing group, there was a statistically significant increase in volume during therapy (*p* < 0.001). The volume change in the control group without stenoses before treatment was not significant.

On average, 62.2% of before and after CBCT evaluations of aligner treatments showed a change in tongue position and its relationship to incisors and palate.

## Figures and Tables

**Figure 1 diagnostics-12-02201-f001:**
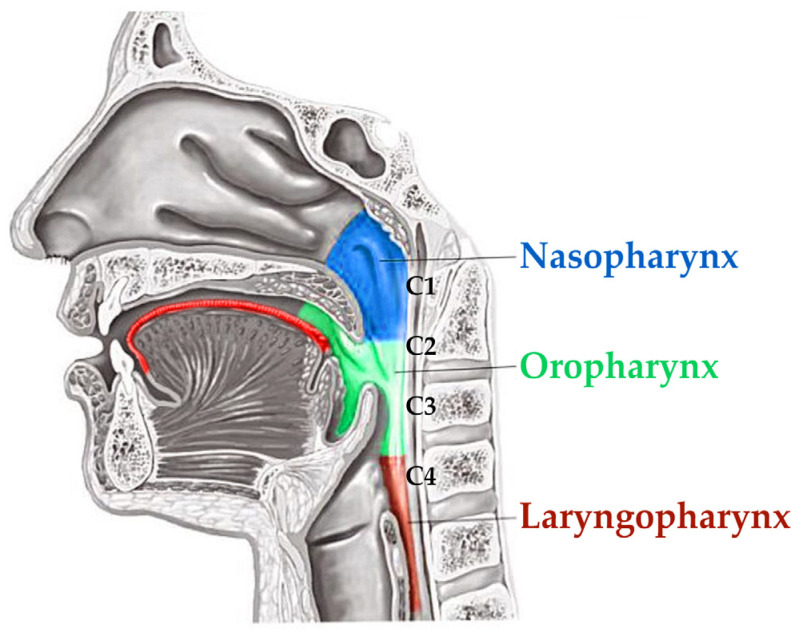
Schematic drawing of tongue position in the oral cavity and oropharynx portion of the pharynx where the airway changes after aligner therapy was evaluated.

**Figure 2 diagnostics-12-02201-f002:**
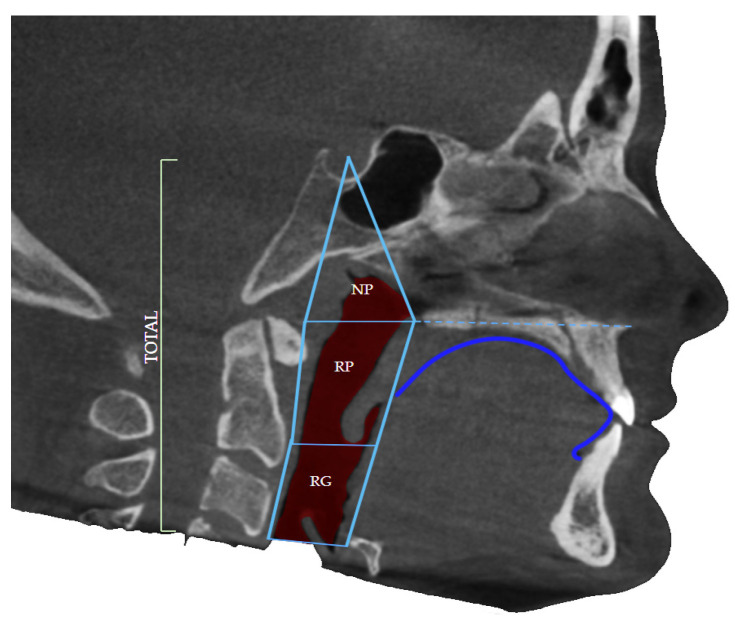
Defined borders of the airway volume consist of the nasopharyngeal (NP) portion as well as oropharyngeal regions: retropalatal (RP) and retroglossal (RG) and dorsum linguae identification.

**Figure 3 diagnostics-12-02201-f003:**
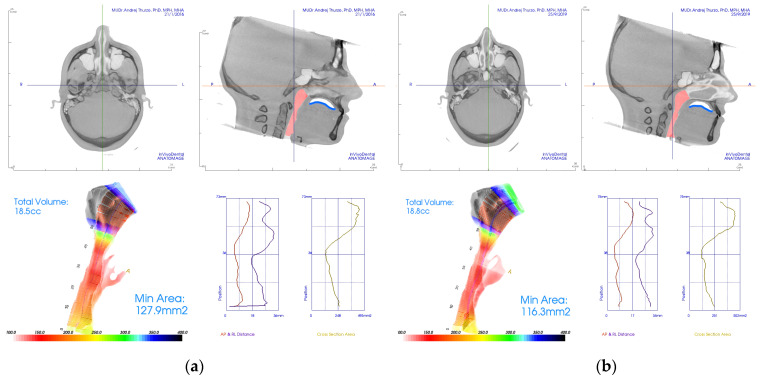
Examples of CBCT 3D analysis of the airway with evaluation of constriction in the software Invivo 6, from Anatomage (Santa Lara, CA, USA). (**a**) Pre-treatment CBCT with constriction; (**b**) post-treatment CBCT analysis with insignificantly improved airway constriction and unchanged tongue position.

**Figure 4 diagnostics-12-02201-f004:**
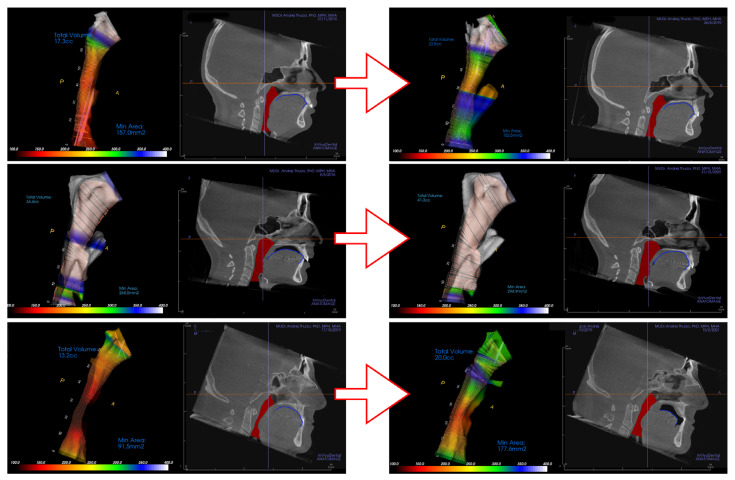
Various clinical situations in airway and tongue position in the oral cavity shown from the top. (1) Initially constricted airway ending in unconstricted setup with the tongue in connection with the upper palate and incisors with obvious change of lower incisors position. (2) In the middle is shown the situation of patient from control group with permanently unconstricted airway with changed tongue position. (3) Example on the bottom shows improvement in the airway clearance despite tongue position in the oral cavity being posterior in comparison to initial CBCT.

**Figure 5 diagnostics-12-02201-f005:**
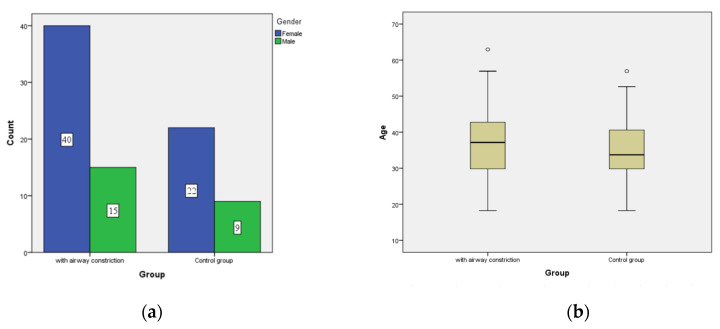
Sex and age distribution between groups. (**a**) The groups did not differ statistically significantly in the proportion of women and men. (**b**) The groups did not differ statistically significantly in the proportion of women and men.

**Figure 6 diagnostics-12-02201-f006:**
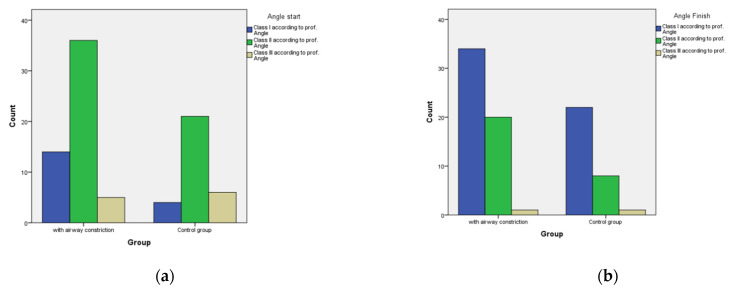
Distribution in Angle classification. (**a**) Initial scan shows Class II as the dominant group in patients with constricted airways as well as in the unconstricted control group. (**b**) At the finishing stage, the majority in both groups were patients in the first Angle Class, although Class II persisted due to strict evaluation of molar relationship, including extraction cases, unilateral Class II or non-surgical camouflage treatments in adults.

**Figure 7 diagnostics-12-02201-f007:**
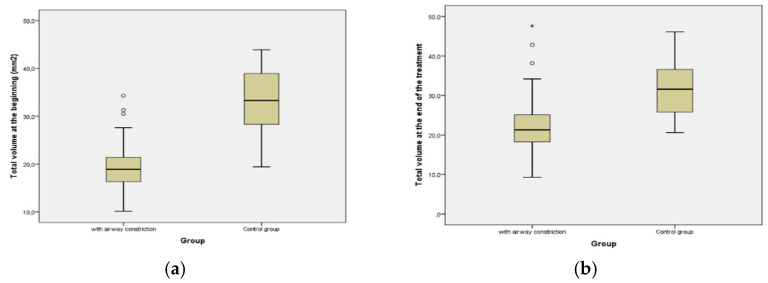
Comparison of volumetric changes of the airway before and after aligner treatment. (**a**) Pre-treatment; (**b**) post-treatment increase of airway volume in Group A and decrease in the control group. Asterisk (*) represents Extreme outlier which is an observation that lies an abnormal distance from other values. Extreme outliers are any data values which lie more than 3.0 times the interquartile range below the first quartile or above the third quartile. Circles (o) represent Common outliers.

**Figure 8 diagnostics-12-02201-f008:**
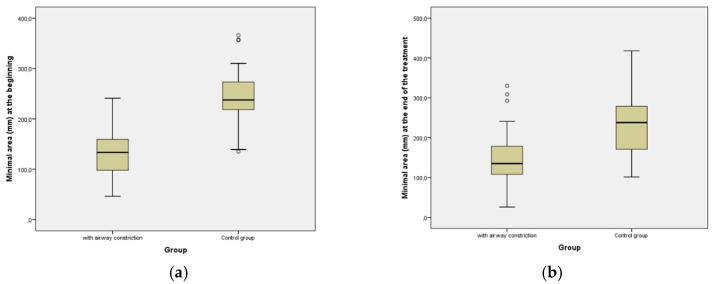
Comparison of changes of the surface of the airway cross-section at the level of highest constriction. (**a**) At the beginning of aligner treatment; (**b**) Post-treatment. The surface of the most constricted cross-section of airway did not change significantly after treatment in any of the groups.

**Figure 9 diagnostics-12-02201-f009:**
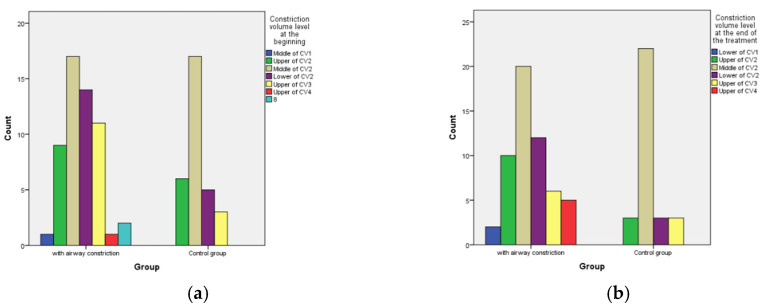
Level of airway constriction according to the level of the cervical vertebrae as shown at the cephalometric lateral X-ray exported from the CBCT. (**a**) Pre-treatment levels of constriction; (**b**) post-treatment levels of highest constriction (smallest clearance).

**Figure 10 diagnostics-12-02201-f010:**
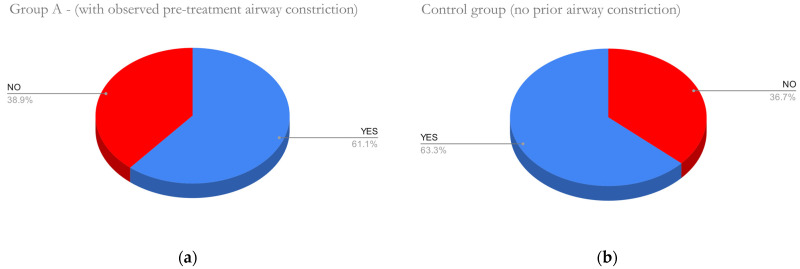
No significant difference between groups was observed. Difference of tongue contour towards palate and incisors between pre-treatment and post-treatment CBCT was assessed. (**a**) In Group A with significant pre-treatment airway constriction, a change of tongue position was observed in 61.1% of cases (**b**) Similarly, 63.3% cases of the control group had a change in tongue position after treatment.

**Figure 11 diagnostics-12-02201-f011:**
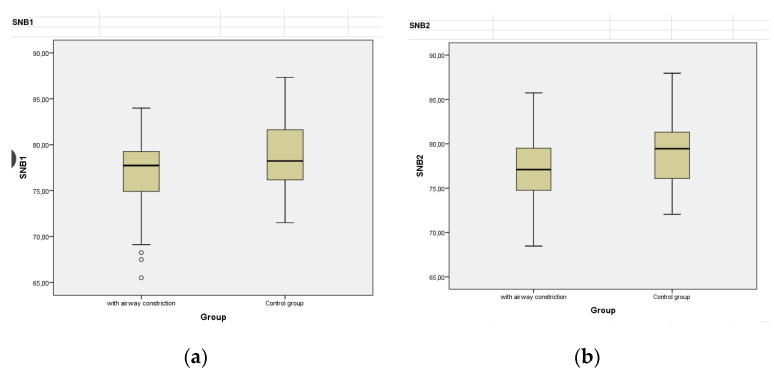
Values of SNB angle in groups A and B. The angle SNB formed by the points (1) Sella Turcica, (2) hard tissue Nasion (3) point B. (**a**) Pre-treatment SNB angle in group A (**left**) and group B (**right**); (**b**) Post-treatment SNB angle in group A (**left**) and group B (**right**).

**Table 1 diagnostics-12-02201-t001:** Definition of cephalometric parameters and their significance.

Parameter	Definition	Significance
SNA	Angle formed by Sella turcica (S), Nasion (N), point A (point of maximum convexity of bony maxilla)	Position of maxilla regarding the skeletal cranial base
SNB	Angle formed by Sella turcica (S), Nasion (N), point B (point of maximum convexity of mandibular profile)	Position of mandible regarding the skeletal cranial base
ANB	Angle formed by point A, Nasion and point B	Angular relationship between the position of the maxilla and the mandible
U1- SN	Inclination of upper central incisors to line connecting Sella and Nasion	Angular change of the inclination of upper incisors regarding the base of the skull
L1-NB	Inclination of the lower incisors to Nasion and B point line	Body protrusion of the lower incisors
OJ	Overjet	Horizontal overlap of upper incisors over the mandibular central incisors
PP-MP or NL-ML	Palatal plane or Nasal line to the mandibular plane/mandibular line	Inclination of mandible in relation to maxilla. Vertical parameter.
MP-SN	Mandibular Plane to Sella- Nasion line	Inclination of mandible in the base of the skull. Rotation of mandible. Vertical parameter.
Gonial Angle	Tangent to the posterior border of the ramus and tangent to the lower border of the mandible	Rotation of the mandible

**Table 2 diagnostics-12-02201-t002:** Inclination of the lower central incisors, test of normality.

	Statistic	df	Significance Correction
L1NB1 Group A L1NB1Group B (Kolmogorov-Smirnova)	,101	55	,200
	,130	31	,199

L1NB2 Group A L1NB2 Group B (Kolmogorov-Smirnova)	,090	55	,200
	,092	31	,200

L1NB1 Group A L1NB1 Group B (Shapiro-Wilk)	,943	55	,012
	,964	31	,380

L1NB2 Group A L1NB2 Group B (Sapiro-Wilk)	,969	55	,165
	,961	31	,313


**Table 3 diagnostics-12-02201-t003:** Comparison between initial and final overjet of t Group A.

	Overjet 1	Overjet 2
N Valid Missing		
	55	55
	0	0
Mean	4.0113	3.5545
Minimum	–3.18	1.89
Maximum	11.29	6.65

## Data Availability

Anonymized Data supporting the reported results are freely available at: https://docs.google.com/spreadsheets/d/1Niq3Yt5SYumA4EQh-ZMq9F19AssL6CvhRBqfcDgI8iY/edit?usp=sharing (accessed on 24 July 2022). Authors ensured that data shared is in accordance with consent provided by participants on the use of confidential data.

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
