# Peer review of "Change in the Constricted Airway in Patients after Clear Aligner Treatment: A Retrospective Study"

_diagnostics, 2022, doi:10.3390/diagnostics12092201_

Round 1
Reviewer 1 Report
Dear Editor,
I would like to thank you the opportunity to review the manuscript entitled “Change in the constricted airway in patients after clear aligner treatment: retrospective study” and so contribute with the Journal.
The present manuscript aimed to present a concept and method of 3D evaluation of upper airway morphology and tongue position from initial and final CBCT accompanying the orthodontic treatment with clear aligners. The subject studied is relevant and of interest to readers. The article is well structured, and its results offer a scientific contribution. Small modifications are necessary so that the article can have the indication of its publication.
See bellow my suggestions:
1. Abstract
- I missed the conclusion.
2. Introduction
- The intro is long. The authors could have been more direct, contextualizing the reader with the topic studied. Another aspect that caught my attention is the high number of claims made without proper scientific support. Example: Page 4, lines 184-186: “During facial growth alterations in upper airway breathing, could affect the stomatognathic system and the development of the structures as well as their functions.”
3. Materials and methods
- Was the sample calculation performed? Why 86 participants?
- Have the examiners been calibrated? If so, how did the calibration process take place?
- What was the level of intra- and inter-examiner agreement? How was this evaluated?
4. Discussion
- The authors could have ended the discussion by presenting the future perspectives opened by carrying out this study.
5. Conclusion
- The conclusion must be redone. Authors should only respond to the objectives presented.
Author Response
Review Report Form
Open Review
|
Yes |
Can be improved |
Must be improved |
Not applicable |
|
|
Does the introduction provide sufficient background and include all relevant references? |
(x) |
( ) |
( ) |
( ) |
|
Are all the cited references relevant to the research? |
(x) |
( ) |
( ) |
( ) |
|
Is the research design appropriate? |
( ) |
(x) |
( ) |
( ) |
|
Are the methods adequately described? |
( ) |
(x) |
( ) |
( ) |
|
Are the results clearly presented? |
(x) |
( ) |
( ) |
( ) |
|
Are the conclusions supported by the results? |
( ) |
(x) |
( ) |
( ) |
Comments and Suggestions for Authors
Response to Reviewer 1 Comments
Dear rewiewer,
Thank you for your positive feedback and recommendations. We have made major revisions to the manuscript text including full syntax revision by native speaker and we have addressed all your remarks. On each of your points we respond below.
Point 1: Abstract:
- I missed the conclusion.
Response 1: Dear colleague, thank you your remark. We have we have reworded conclusions in the abstract.
Point 2: Introduction
- The intro is long. The authors could have been more direct, contextualizing the reader with the topic studied. Another aspect that caught my attention is the high number of claims made without proper scientific support. Example: Page 4, lines 184-186: “During facial growth alterations in upper airway breathing, could affect the stomatognathic system and the development of the structures as well as their functions.”
Response 2: The introduction has been reorganized and reduced as possible. The requested part of research that affects the stomatognathic system is based on references (20, 21). Other statements and references were revised.
Point 3: Materials and methods
- Was the sample calculation performed? Why 86 participants? - Have the examiners been calibrated? If so, how did the calibration process take place? - What was the level of intra- and inter-examiner agreement? How was this evaluated?
Response 3:
This is a retrospective study and all participants meeting inclusion criteria were included. Full statistics of sample calculation is provided (line 848-852) in Data Availability Statement: Anonymized Data supporting the reported results are freely available at:https://docs.google.com/spreadsheets/d/1Niq3Yt5SYumA4EQh-ZMq9F19AssL6CvhRBqfcDgI8iY/edit?usp=sharing.Authors ensured that data shared is in accordance with consent provided by participants on the use of confidential data. An extensive pool of patients was searched for our inclusive criteria. For group A fifty-five patients were fullfiling the defined parameters. That means initial and final CBCT and upper airway constriction shown in their initial CBCTs. About control group a sufficient amount of thirty-one patients were selected.
In regard to Examiner Standardization and Calibration: this is necessary in clinical research that typically relies on clinical examiners to assess variables. In this paper, all major objectives were assesed by specialized software Invivo 6 from Anatomage. The only task where the examiners were evaluating subjectively was the very simple binary task of tongue position change. Examiners agreement in Group A was 54/55 (98.20%) and in the Control group was 31/31 (100%). In the only case of initial disagreement a mutual agreement was reached. Intra-rater reliability was estimated by having the rater read and evaluate each tongue position 2 times in two separate dates. The intra examiner reproducibility was 100% in this simple task. Interexaminer reliability—reaching agreement between two examiners—was typically more tedious, albeit only one case of disagreement occurred. For expressing interexaminer reliability is the kappa statistic with examiners calibration protocols was considered not necessary in this case in wider extend than already was published in statistical evaluation in the provided statistical data output available> https://docs.google.com/spreadsheets/d/1Niq3Yt5SYumA4EQh-ZMq9F19AssL6CvhRBqfcDgI8iY/edit?usp=sharing.
We have added the explanation into the text in the Methods chapter.
Point 4: Discussion
- The authors could have ended the discussion by presenting the future perspectives opened by carrying out this study.
Response 4: Thank you for your suggestion. We have reduced the Discusion chapter and removed some references.
Point 5: Conclusion
- The conclusion must be redone. Authors should only respond to the objectives presented.
Response 5: We had reformulated the conclusions.
Kind regards Dr. Fountoulaki and Thurzo

Reviewer 2 Report
Dear authors.
First of all, thank you for giving me the chance to review your article
The introduction is too long. The authors should be concise and avoid excessive references.
References: old and too specific for brain diseases
Reference 29: This is a review study. Please cite the original study about the information of the Italian team that developed the respiratory test.
There is a review of the literature in the methods section, and it sounds strange to the readers to discuss the methods of other studies in the methodology section.
The IRB consent for this study is not informed. studies using CBCT scans must take care of the time interval, in order to avoid biological risks for the patients. Moreover, the interval between the first and the second scan is missing.
Sample:
The sample is too heterogeneous. Patients submitted to orthodontic comprehensive treatment and orthognathic surgery treatment have different results on airway volume before and after treatment. The type of surgery can also influence the results.
Should be paired with the type of treatment (with or without teeth extraction), and exclude patients with mandibular advancement surgeries.
The image analysis must contain important information, such as calibration and reliability values.
I’m sorry. The study must be better designed for its objectives. The samples show an important bias. I suggest to the authors think more about the methods.
Author Response
Open Review
(x) Extensive editing of English language and style required
( ) Moderate English changes required
( ) English language and style are fine/minor spell check required
( ) I don't feel qualified to judge about the English language and style
|
Yes |
Can be improved |
Must be improved |
Not applicable |
|
|
Does the introduction provide sufficient background and include all relevant references? |
( ) |
( ) |
(x) |
( ) |
|
Are all the cited references relevant to the research? |
( ) |
( ) |
( ) |
(x) |
|
Is the research design appropriate? |
( ) |
( ) |
( ) |
(x) |
|
Are the methods adequately described? |
( ) |
( ) |
( ) |
(x) |
|
Are the results clearly presented? |
( ) |
( ) |
( ) |
(x) |
|
Are the conclusions supported by the results? |
( ) |
( ) |
( ) |
(x) |
Response to Reviewer 2 Comments
Dear colleague,
Thank you for your feedback and recommendations. As you have recommended to perform Extensive editing of English language and style, we have made major revisions to the manuscript text including full syntax revision by native speaker and we have addressed your remarks. On each of your points we respond below.
Point 1: The introduction is too long. The authors should be concise and avoid excessive references.
Response 1: The manuscript has been reorganized and the introduction has been reduced.
Point 2: References: old and too specific for brain diseases Reference 29: This is a review study. Please cite the original study about the information of the Italian team that developed the respiratory test.
Response 2:
-References have been selected wisely and carefully to evidence-base our project. Since our topic is highly linked also to sleep mechanism we based our research on two references of year 2021, one of 2018, one of 2009 and two fundamental ones from year 1998.
-Original study has been correctly cited now, by spanish team of leading Dr. Fortuna.
Point 3: There is a review of the literature in the methods section, and it sounds strange to the readers to discuss the methods of other studies in the methodology section.
Response 3: The study of dos Santos et al in 2020 was mentioned in the methodology section since their study gave inspiration to define the borders during our airway measurements. They used similar Imaging software (Dolphin Imaging) like our team did and we found their study very reliable and worth mentioning.
Point 4: The IRB consent for this study is not informed. studies using CBCT scans must take care of the time interval, in order to avoid biological risks for the patients. Moreover, the interval between the first and the second scan is missing.
Response 4: Dear reviewer, this is a retrospective study, and the CBCT examination was not performed for the benefits of the research and naturally the time interval between scans varies depending on many variables including treatment length. IRB consent is not applicable. In this research the researchers conducted data collection that was subject to the General Data Protection Regulation (GDPR) so a general patient consent has been provided and signed as part of the treatment. The mean duration of the treatment was two-years and is defined in detail in in the group selection and clinical evaluation chapter 2.3.
Point 5:
Sample: The sample is too heterogeneous. Patients submitted to orthodontic comprehensive treatment and orthognathic surgery treatment have different results on airway volume before and after treatment. The type of surgery can also influence the results.
Should be paired with the type of treatment (with or without teeth extraction), and exclude patients with mandibular advancement surgeries.
Response 5: Dear rewiewer, your point is correct and thank you for your remark. Let us explain our considerations and deccisions. All selected cases were non-extraction treatments (Line 554). We have spent lots of considerations in regards to selection criteria. The full statistical analysis was provided with the manuscript and as you can see in data sample analysis with statistical evaluation (in Data Availability Statement: Anonymized Data supporting the reported results are available at: https://docs.google.com/spreadsheets/d/1Niq3Yt5SYumA4EQh-ZMq9F19AssL6CvhRBqfcDgI8iY/edit?usp=sharing) – only one type of surgeries is included (Only Class II – mandibular advancement mono-maxillary surgeries were in the sample). Upon research of the dataset we came to consensus that to reach less heterogenity of the sample is nearly impossible and even if suceeded it would render the output of the manuscript extremly biased and statistically weak. As it is important to admit that classification by Angle Classes does not determine the extend of the treatment. Variability of malocclusions is so extensive, that concentration in effort to narrow the dataset by type of maloclusion, or used treatment modality would result in even more bias. For example use of elastics can vary in length, forse, extend of the teeth movement and is nearly impossible to responsibily approximate and preserve statistically relevant sample. Both of your points: 1. Extractions and 2.surgery type are addresed. No extraction cases were in the evaluated sample and only Only Class II – mandibular advancement mono-maxillary surgeries were in the sample.This is now clarified in the text. Our decision not to remove these surgical cases was based upon statistical evaluation. Occurrence of surgical AII cases was not different between groups, albeit it initially seemed logical that Class II surgical cases will have constricted airway and will not occur in the control group. 5.5% of group A(constricted) and 3.2% of group B (control) underwent orthognathic intervention - Class II – mandibular advancement mono-maxillary surgery. Also the statistical evaluation (shown in the online dataset) showed no statisticaly significant difference between groups. For these reasons we see no benefit, and prefer not to remove treatments with surgical intervention from the sample.
Point 6: The image analysis must contain important information, such as calibration and reliability values.
Response 6: Dear reviewer, thank you for another valuable remark. In regard to Examiner Standardization and Calibration: this is necessary in clinical research that typically relies on clinical examiners to assess variables. In this paper, all major objectives were assesed by specialized software Invivo 6 from Anatomage. The only task where the examiners were evaluating subjectively was the very simple binary task of tongue position change. Examiners agreement in Group A was 54/55 (98.20%) and in the Control group was 31/31 (100%). In the only case of initial disagreement a mutual agreement was reached. Intra-rater reliability was estimated by having the rater read and evaluate each tongue position 2 times in two separate dates. The intra examiner reproducibility was 100% in this simple task. Interexaminer reliability—reaching agreement between two examiners—was typically more tedious, albeit only one case of disagreement occurred. For expressing interexaminer reliability is the kappa statistic with examiners calibration protocols was considered not necessary in this case in wider extend than already was published in statistical evaluation in the provided.
We have added the explanation into the text in the Methods chapter. Lines 384-398
Kind regards Dr. Fountoulaki and Thurzo

Reviewer 3 Report
Reviewer's comment for Diagnostics-1864462
Comments to the Author
This manuscript is about: Change in the Constricted Airway in Patients After Clear Aligner Treatment: Retrospective Study.
It's an interesting topic and the purpose of this study was to compares the clinical effect of aligner therapy on upper airway morphology in patients with constriction prior to the treatment and patients without constriction.
1. The context of this manuscript seemed to be abundant, but this is not a master’s thesis or a doctoral dissertation. It is recommended to re-submit after reorganizing the data.
2. English should be edited again by a native English teacher.
(eg.) The grammar is not correct such as on page 5of 22 (From line215-222)
3. The content of [Material and Methods] (From line 377-393) was the definition of cephalometric parameters. Please express the content in the form of Table.
4. The resolution of figure. 2, 3, 4, 5, 6, 7, 8, 9, 11 were not fair. The number of figures was too much. please re-organized to the format of tables.
Author Response
Open Review
(x) Extensive editing of English language and style required
( ) Moderate English changes required
( ) English language and style are fine/minor spell check required
( ) I don't feel qualified to judge about the English language and style
|
Yes |
Can be improved |
Must be improved |
Not applicable |
|
|
Does the introduction provide sufficient background and include all relevant references? |
(x) |
( ) |
( ) |
( ) |
|
Are all the cited references relevant to the research? |
( ) |
(x) |
( ) |
( ) |
|
Is the research design appropriate? |
(x) |
( ) |
( ) |
( ) |
|
Are the methods adequately described? |
(x) |
( ) |
( ) |
( ) |
|
Are the results clearly presented? |
(x) |
( ) |
( ) |
( ) |
|
Are the conclusions supported by the results? |
(x) |
( ) |
( ) |
( ) |
Response to Reviewer 3 Comments
Dear colleague,
Thank you for your feedback and recommendations. As you have recommended to perform Extensive editing of English language and style, we have made major revisions to the manuscript text including full syntax revision by native speaker and we have addressed your remarks. On each of your points we respond below.
Point 1: The context of this manuscript seemed to be abundant, but this is not a master’s thesis or a doctoral dissertation. It is recommended to re-submit after reorganizing the data.
Response 1: The manuscript has been reduced in length and has been reorganized, containing actual useful data to allow understanding of the topic. The length has been reduced as possible.
Point 2: English should be edited again by a native English teacher.
(eg.) The grammar is not correct such as on page 5of 22 (From line215-222)
Response 2: A native speaker controlled our manuscript and made their suggestions which we have implemented with more than 50 syntax corrections.
Point 3: The content of [Material and Methods] (From line 377-393) was the definition of cephalometric parameters. Please express the content in the form of Table.
Response 3: The cephalometric parameters that were taken into account are provided in the following Table form.
Table 1. Cephalometric skeletal parameters and explanation of their content.
|
Parameter |
Definition |
Significance |
|
SNA |
Angle formed by Sella turcica (S), Nasion (N), point A (point of maximum convexity of bony maxilla) |
Position of maxilla regarding the skeletal cranial base |
|
SNB |
Angle formed by Sella turcica (S), Nasion (N), point B (point of maximum convexity of mandibular profile) |
Position of mandible regarding the skeletal cranial base |
|
ANB |
Angle formed by point A, Nasion and point B |
Angular relationship between the position of the maxilla and the mandible |
|
U1- SN |
Inclination of upper central incisors to line connecting Sella and Nasion |
Angular change of the invlination of upper incisors regarding the base of the skull |
|
L1-NB |
Inclination of the lower incisors to Nasion and B point line |
Body protrusion of the lower incisors |
|
OJ |
Overjet |
Horizontal overlap of upper incisors over the mandibular central incisors |
|
PP-MP or NL-ML |
Palatal plane or Nasal line to the mandibular plane/mandibular line |
Inclination of mandible in relation to maxilla. Vertical parameter. |
|
MP-SN |
Mandibular Plane to Sella- Nasion line |
Inclination of mandible in the base of the skull. Rotation of mandible. Vertical parameter. |
|
Gonial Angle |
Tangent to the posterior border of the ramus and tangent to the lower border of the mandible |
Rotation of the mandible |
Point 4: The resolution of figure. 2, 3, 4, 5, 6, 7, 8, 9, 11 were not fair. The number of figures was too much. please re-organized to the format of tables.
Response 4: Dear reviewer, the figures provided in the text/PDF are intentionally reduced for complete file size. The high-definition images in PSD photoshop format as well as PNG are provided to editors separately, albeit thank you for your observation.
Kind regards Dr. Fountoulaki and Thurzo

Round 2
Reviewer 2 Report
There is no information about the validation of the methodology of tongue evaluation from previous studies. Is this method accurate? Moreover, the tongue position evaluation cannot be considered an effect of the treatment using this methodology. This assessment has some bias, due the possibility of tongue movement during the scanning, even with no swallow, as seen in Figure 4. The inferior image shows a gap between the tongue and the hard palate after treatment. For other hand, the pretreatment tongue position is not resting on the hard palate in the second image of Figure 4. In my opinion, the patients should be evaluated clinically before the tomography scanning about the tongue position on resting.
There is a long discussion of the method in methods section. This information is better suitable in discussion section.
“Research from 2016 published by Pliska et al. in American Journal of Orthodontics 276 and Dentofacial Orthopedics was focused on the effect of orthodontic treatment on the 277 upper airway volume in adults. In this retrospective study authors have examined the 278 effects of orthodontic treatment with and without extractions on the airway clearance. ..., hypopharyngeal, and hyoid position in bimaxillary 290 protrusive adult patients” [49].”
The sample is too heterogeneous, with different types of treatment and orthodontic mechanics, including orthognathic surgeries. For the reader it will be hard to understand which aspect can influence more the airway changes. This weakness of the sample selection is too important to validate the results. For this reason, this study should be better designed about the sample.
Reviewer 3 Report
Dear Authors:
All the recommendations are responded to, this article is suitable to be accepted in the present form.
Best regards!
Author Response
Dear reviewer,
thank you for your approval.
Upon request of the editor, we have further reduced the Introduction, elaborated on the reasons the IRB consent was waived, added clinical heterogeneity as a limitation and improved some english syntax.
kind regards